# Expression of the cancer-associated DNA polymerase ε P286R in fission yeast leads to translesion synthesis polymerase dependent hypermutation and defective DNA replication

Ignacio Soriano[1,2]*, Enrique Vazquez[3], Nagore De Leon[1,2°], Sibyl Bertrand[1°], Ellen Heitzer[4], Sophia Toumazou[1,5], Zhihan Bo[1], Claire Palles[6], Chen-Chun Pai[5], Timothy C. Humphrey[5], Ian Tomlinson[2], Sue Cotterill[7], Stephen E. Kearsey[1]*

1 ZRAB, University of Oxford, Oxford, United Kingdom, 2 Edinburgh Cancer Research Centre, Institute of Genetics and Cancer, University of Edinburgh, Western General Hospital, Edinburgh, United Kingdom, 3 Genomics Unit, Centro Nacional de Investigaciones Cardiovasculares (CNIC), Madrid, Spain, 4 Institute of Human Genetics, Diagnostic & Research Center for Molecular BioMedicine, Medical University of Graz, Graz, Austria, 5 MRC Oxford Institute for Radiation Oncology, Department of Oncology, University of Oxford, Oxford, United Kingdom, 6 Gastrointestinal Cancer Genetics Laboratory, Institute of Cancer and Genomic Sciences, College of Medical and Dental Sciences, University of Birmingham, Birmingham, United Kingdom, 7 St. George's, University of London, Cranmer Terrace, Tooting, London, United Kingdom

☯ These authors contributed equally to this work.
* ignacio.soriano@igmm.ed.ac.uk (IS); stephen.kearsey@zoo.ox.ac.uk (SEK)

**Data Availability Statement:** Sequencing data from this study have been deposited in the Gene

## Abstract

Somatic and germline mutations in the proofreading domain of the replicative DNA polymerase ε (*POLE*-exonuclease domain mutations, *POLE*-EDMs) are frequently found in colorectal and endometrial cancers and, occasionally, in other tumours. POLE-associated cancers typically display hypermutation, and a unique mutational signature, with a predominance of C > A transversions in the context TCT and C > T transitions in the context TCG. To understand better the contribution of hypermutagenesis to tumour development, we have modelled the most recurrent *POLE*-EDM (*POLE-P286R*) in *Schizosaccharomyces pombe*. Whole-genome sequencing analysis revealed that the corresponding *pol2-P287R* allele also has a strong mutator effect in vivo, with a high frequency of base substitutions and relatively few indel mutations. The mutations are equally distributed across different genomic regions, but in the immediate vicinity there is an asymmetry in AT frequency. The most abundant base-pair changes are TCT > TAT transversions and, in contrast to human mutations, TCG > TTG transitions are not elevated, likely due to the absence of cytosine methylation in fission yeast. The *pol2-P287R* variant has an increased sensitivity to elevated dNTP levels and DNA damaging agents, and shows reduced viability on depletion of the Pfh1 helicase. In addition, S phase is aberrant and RPA foci are elevated, suggestive of ssDNA or DNA damage, and the *pol2-P287R* mutation is synthetically lethal with *rad3* inactivation, indicative of checkpoint activation. Significantly, deletion of genes encoding some translesion synthesis polymerases, most notably Pol κ, partially suppresses *pol2-P287R* hypermutation, indicating that polymerase switching contributes to this phenotype.

Expression Omnibus (GEO) database under the accession number GSE169231 (https://www.ncbi.nlm.nih.gov/geo/query/acc.cgi?acc=GSE169231).

**Funding:** This work was funded by the Medical Research Council (grant MR/L016591/1 to S.E.K. and S.C.) and the European Research Council (EVOCAN grant 340560 to I.T.). The funders had no role in study design, data collection and analysis, decision to publish, or preparation of the manuscript.

**Competing interests:** The authors have declared that no competing interests exist.

## Author summary

Cancer is a genetic disease caused by mutations that lead to uncontrolled cell proliferation and other tumour properties. Defects in DNA repair or replication can lead to cancer development by increasing the likelihood that cancer-causing mutations will happen. Here we look at a pathogenic variant of a polymerase involved in genome replication (DNA polymerase ε POLE-P286R). This variant is associated with highly mutated cancer genomes. By introducing this mutation into the polymerase ε gene of a model organism, fission yeast, we show that it causes a large increase in single base substitutions, scattered throughout the genome. The sequence context of mutations is similar in fission yeast and humans, suggesting that the yeast model is useful for understanding how POLE-P286R causes such a high mutation rate. Yeast POLE-P286R cells show slow chromosome replication, suggesting that the polymerase has difficulty in copying certain chromosomal regions. Yeast POLE-P286R cells become inviable when the concentration of dNTP building blocks for DNA synthesis is increased, probably because the mutation rate is pushed to an intolerable level. Interestingly, we find that specialised polymerases that are tolerant of DNA damage contribute to the high mutation rate caused by POLE-P286R. These findings have implications for the therapy of POLE-P286R tumours.

## Introduction

In eukaryotes, nuclear DNA replication is carried out by three members of the B-family of DNA polymerases (Pols), Pols α, δ and ε, which function cooperatively to guarantee accurate and efficient genome duplication. Pol α synthesizes the primer to initiate DNA replication, allowing Pol ε and Pol δ to take over leading and lagging strand synthesis, respectively. Unlike Pol α, Pols ε and δ display high processivity and fidelity, being the only nuclear polymerases with functional 3'-5' exonuclease activity capable of correcting mistakes made during DNA synthesis [1,2].

Pol ε is a large, four-subunit protein with critical roles in DNA replication and repair, cell cycle control and epigenetic inheritance (reviewed in [3]). In contrast to Pol δ, Pol ε is a highly processive enzyme even in the absence of accessory factors [4–6], and is perhaps the most accurate eukaryotic DNA polymerase [7,8]. The high intrinsic processivity is due to the presence of a small domain in the catalytic subunit (Pol2) that allows Pol ε to encircle the nascent dsDNA [6,9]. Another unique feature of Pol2 is the presence of a tyrosine (Y431 in *S. cerevisiae*) in the major groove of the nascent base-pair binding pocket, which may contribute to the fidelity of polymerisation [9].

Proofreading increases Pol ε replication accuracy by ~100-fold [7] and has an essential role in the maintenance of genomic stability. Mutations inactivating the exonuclease activity of Pol ε cause an increased mutation rate in both yeast [10] and mice, and lead to murine tumours in tissues with a high rate of cell turnover [11]. Recently, these findings were shown to be relevant to human cancer from large-scale studies of colorectal (CRC) and endometrial cancers (EC) [12–15]. This analysis revealed a subset of hypermutated microsatellite-stable (MSS) tumours with heterozygous somatic mutations in the exonuclease domain of Pol ε. A further study of sporadic ECs showed that somatic *Pol ε* exonuclease domain mutations (*POLE* EDMs) were present in about 7% of cases [16]. Subsequently, several thousand colorectal and endometrial tumour samples have been analysed, reporting more than 10 different *POLE* driver mutations [17,18]. Current data suggest that somatic *POLE* proofreading domain mutations are present

in 1–2% of CRCs and 7–12% of ECs, and less commonly in hypermutated tumours of the brain, pancreas, ovary, breast, stomach, lung and prostate [18,19]. The most striking molecular characteristic in tumours harbouring many somatic *POLE* EDMs is their very high mutation rate, often exceeding 100 mutations/Mb. These mutations are predominantly single base substitutions and define a characteristic mutational pattern with a high proportion of C > A transversions in the context TCT, C > T transitions in the context TCG and T > G transversions in the context TTT, corresponding to COSMIC signatures SBS10a/b [16,20–22]. Recent findings suggest however that the authentic *POLE* EDMs signature is SBS14, and SBS10a/b may result from mismatch repair (MMR)-mediated skewing of SBS14 [23–25].

The amino-acids substituted in somatic *POLE* EDMs show a variable incidence rate (reviewed in [18,19]), with *POLE-P286R* being the most frequent variant in colorectal and endometrial cancer [16,21,26], reaching a frequency of ~7% in early-onset colorectal cancer [27]. Heterozygous *Pole-P286R* mice develop malignant tumours of diverse lineages, which show very high mutation rates in the range of human malignancies [28]. This exceptional mutator phenotype presumably leads to a greater cancer risk due to mutations in driver genes, explaining the frequent occurrence of this variant in cancers.

The equivalent substitution (*pol2-P301R*) has been functionally validated in *S. cerevisiae*, showing an unusually strong mutator effect, and even in the heterozygous state the mutation rate to canavanine resistance is elevated 27 times relative to wild-type [26], comparable with complete MMR deficiency. Interestingly, budding yeast expressing Pol2-P301R shows a mutation rate far higher than that of the exonuclease inactive variant (Pol ε exonull), suggesting that P286R is able to increase the mutation rate through other processes other than proofreading deficiency alone [26,29]. Curiously, the mutation rate of Pol2-P301R *in vitro* is less than that of Pol ε exonull, and is reported to have increased polymerase activity with the ability to extend mismatched primer termini. It is possible that this phenotype may result from the P286R mutation blocking the nascent DNA terminus from switching to the exonuclease site [30]. Nevertheless, the *S. cerevisiae* Pol2-P301R holoenzyme still shows some exonuclease activity [29], while the human enzyme has been reported to have none [21].

Defects in polymerases may lead to increases in mutation rates by several mechanisms. In *S. cerevisiae*, an exonuclease null Pol δ mutant activates mutagenic repair via a checkpoint pathway that elevates dNTP levels, and this may be more important for accumulation of mutations than the proofreading defect *per se* [31]. The mechanism behind the mutator phenotype of the variant *POLD1-R689W* is similar in that yeast cells expressing Pol3-R696W (equivalent to *POLD1-R689W*) show a checkpoint-dependent increase in dNTP levels, similar to that seen with proofreading defective mutants [32]. This elevation of dNTP pools further increases the rate of Pol δ errors, thus forming a vicious circle and a similar mechanism has been suggested to explain the mutator phenotype due to error-prone DNA polymerase ε variants [33].

An additional mechanism linking polymerase defects to an increased mutation rate involves translesion synthesis (TLS) polymerases. In *S. pombe*, Pol κ, and Pol ζ to a lesser extent, contribute significantly to the mutator phenotype of a strain expressing defective Pol α [34]. This study suggests that TLS polymerases are recruited in response to replication fork stalling or collapse, restarting synthesis at the cost of an increased replicative mutation rate. However, Pol ζ is not responsible for the P286R hypermutation phenotype in *S. cerevisiae* [29].

In this study, we have characterized the impact of the P286R variant on genome-wide mutation and S phase execution in the fission yeast *Schizosaccharomyces pombe*. Furthermore, we studied the effect of increasing the mutation frequency in the viability of cells with high mutational burden. Finally, we addressed the contribution of TLS polymerases to hypermutation.

## Results

### Fission yeast mimic of POLE-P286R has a strong mutator phenotype

P286 is invariant in all Pol ε orthologues, and is also conserved in Pol δ, T4 and RB69 bacterio-phage polymerases (Fig 1A). A fission yeast mimic of *POLE-P286R* (*pol2-P287R*) and an exo-nuclease deficient variant (*pol2-D276A/E278A*, [35,36]) were constructed and *in vivo* mutation rates were measured by fluctuation analysis. Introduction of these mutations did not affect the steady state levels of the protein (S1 Fig). The Pol2-P287R variant is also hypermutagenic in fis-sion yeast, with a rate of mutation that is about 70 times higher than wild type and about 5-fold higher than the catalytically dead mutant (Fig 1B), supporting the hypothesis that this

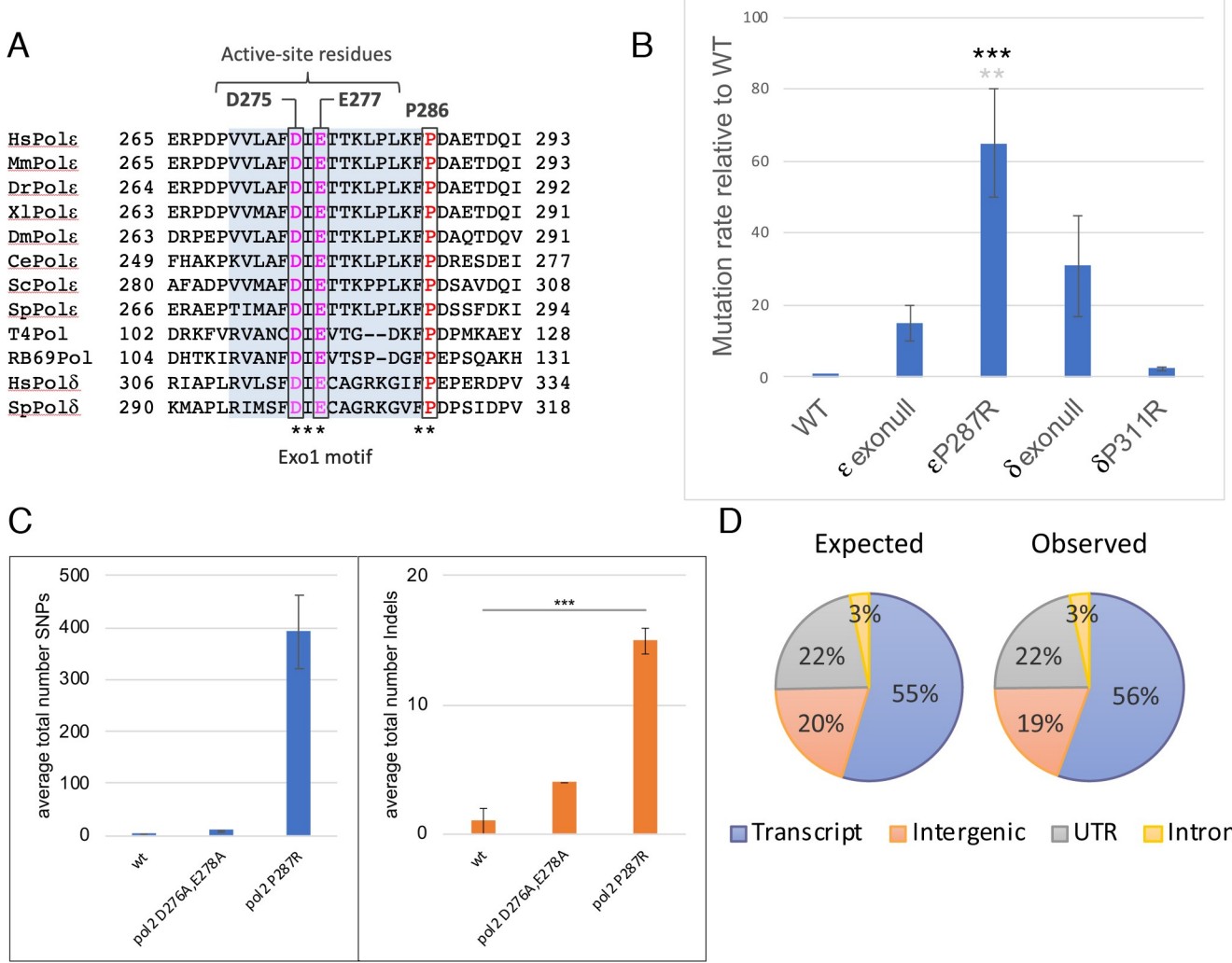

**Fig 1. The *POLE-P286R* hypermutagenic phenotype is conserved in *S. pombe*.** (A) Multiple sequence alignment showing that the Pro286 residue (red) adjacent to the Exo I domain (blue shadow) is absolutely conserved in Pol ε and Pol δ orthologues as well as in T4 and RB69 phage polymerases. Catalytic residues of the 3'->5' exonuclease domain are highlighted in magenta. (B) CanR mutation rates of *pol2-P287R*, *pol3-P311R* (equivalent proline to Pol2-P287), and exonull Pol2 and Pol3 relative to wild type. A multiple comparison using the ordinary one-way ANOVA revealed that the *pol2-P287R* strain has a significantly greater mutation rate compared to wild-type (black asterisks, *** P<0.0001) and *pol2-D276A/E278A* (exonull) (grey asterisks, ** P<0.01). Numerical data are in S13 Table. (C) Number of base substitutions and indels identified in the 200 generation mutation accumulation (MA) experiment in wild-type, exonull and *pol2-P287R* cells. The number of lineages set up was: wt 2; *pol2-D276 E278A* 2; *pol2-P287R* 3. The average number of mutations is shown and error bars show the range of data. ***P<0.001. Numerical data are shown in S10 Table. (D) Expected and observed distribution of mutations obtained in *pol2-P287R* cells during the MA experiment across different genomic regions.

change results in defects substantially more severe than loss of proofreading. Interestingly, mutation of the equivalent proline to arginine (P311R) in *S. pombe* Pol δ (Pol3/Cdc6) did not result in a hypermutating phenotype (Fig 1B), possibly reflecting different consequences of Pol ε and Pol δ malfunction on leading and lagging strands.

To analyse further the rate and the spectrum of mutations, we undertook a ~200-generation mutation accumulation (MA) experiment followed by deep sequencing of genomic DNA from wild-type and mutant cells. On average, we identified 0.5, 9.5 and 394 single-base substitutions in the wild-type, *pol2-D276A/E278A* and *pol2-P287R* strains, respectively (Fig 1C). Consistent with the fluctuation assay data, *pol2-P287R* cells have the highest rate of mutation, exceeding that of *pol2-D276A/E278A* mutant by more than 40 times (Fig 1C). In addition, this hypermutator variant displayed a much smaller but still significant increase in indels (Fig 1C), consonant with the microsatellite-stable (MSS) phenotype of *POLE* EDM tumours [16]. There was no genomic region particularly prone to P287R-induced mutations, which were equally distributed across the genome (Fig 1D).

## Fission yeast mutations induced by *pol2-P287R* broadly recapitulate mutations seen in *POLE-P286R* cancer genomes

We next wished to compare the mutation spectra occurring in *POLE-P286R* human cancers and the equivalent mutant in fission yeast. First, we analysed whole-exome sequencing data from the TCGA colorectal and endometrial tumours harbouring the P286R variant. As expected, both groups of samples displayed very similar patterns, typified by C > A transversions in the context TCT, C > T transitions in the context TCG and T > G transversions in the context TTT (Fig 2A), which is in accordance with the specific mutational spectrum of *POLE* EDMs, although this is after it has been altered by MMR and possibly Pol δ proofreading [16,20,21,37,38].

We then compared the pattern of *pol2-P287R* mutations to the human spectra (Fig 2A and 2B upper panel). These showed a moderate resemblance (cosine similarity 0.68) (S5 Table) and, in consonance with human cancers, the most abundant mutations are C > A transversions in the context TCT, although not to the same proportion as in the tumour genomes (Fig 2A and 2B). In addition, the *pol2-P287R* cells recapitulate the most discriminating genomic alterations for human *POLE* mutations: CG to AT ≥ 20%; TA to GC ≥ 4%; CG to GC ≤ 0.6% [39]. More precisely, *pol2-P287R* cells present values of 33%, 15%, and 0% respectively (Figs 2B and S2). However, the lack of TCG > TTG transitions in the *pol2-P287R* spectrum is noteworthy. In humans, C > T transitions in a CpG base context are probably due to the deamination of 5-methylcytosine to thymine [40,41] and/or mutant POLE synthesis across methylated cytosines situated on the leading strand [42], and the absence of 5-methylcytosine in fission yeast may explain these differences. Overall, these data suggest that fission yeast expressing Pol2-P287R is a useful model for the human mutation.

Genome-wide estimates of mutation rates and mutational spectrum of wild-type *S. pombe* have been recently published [43,44]. We used the base-substitution mutations from both studies to generate the mutational spectrum of the wild-type strain and we then compared it with the percentage of base substitutions identified in *pol2-P287R* cells (Fig 2B). We found a prominent enrichment of C > A transversions in an NCT context and T > G transversions in an NTT context in the *pol2-P287R* mutant. In contrast, C > G transversions occurred with increased frequency in the wild-type cells, being completely absent in the mutant strain (Fig 2B). This indicates that the Pol2-P287R polymerase is not simply amplifying the pattern of mutations found in wild-type cells.

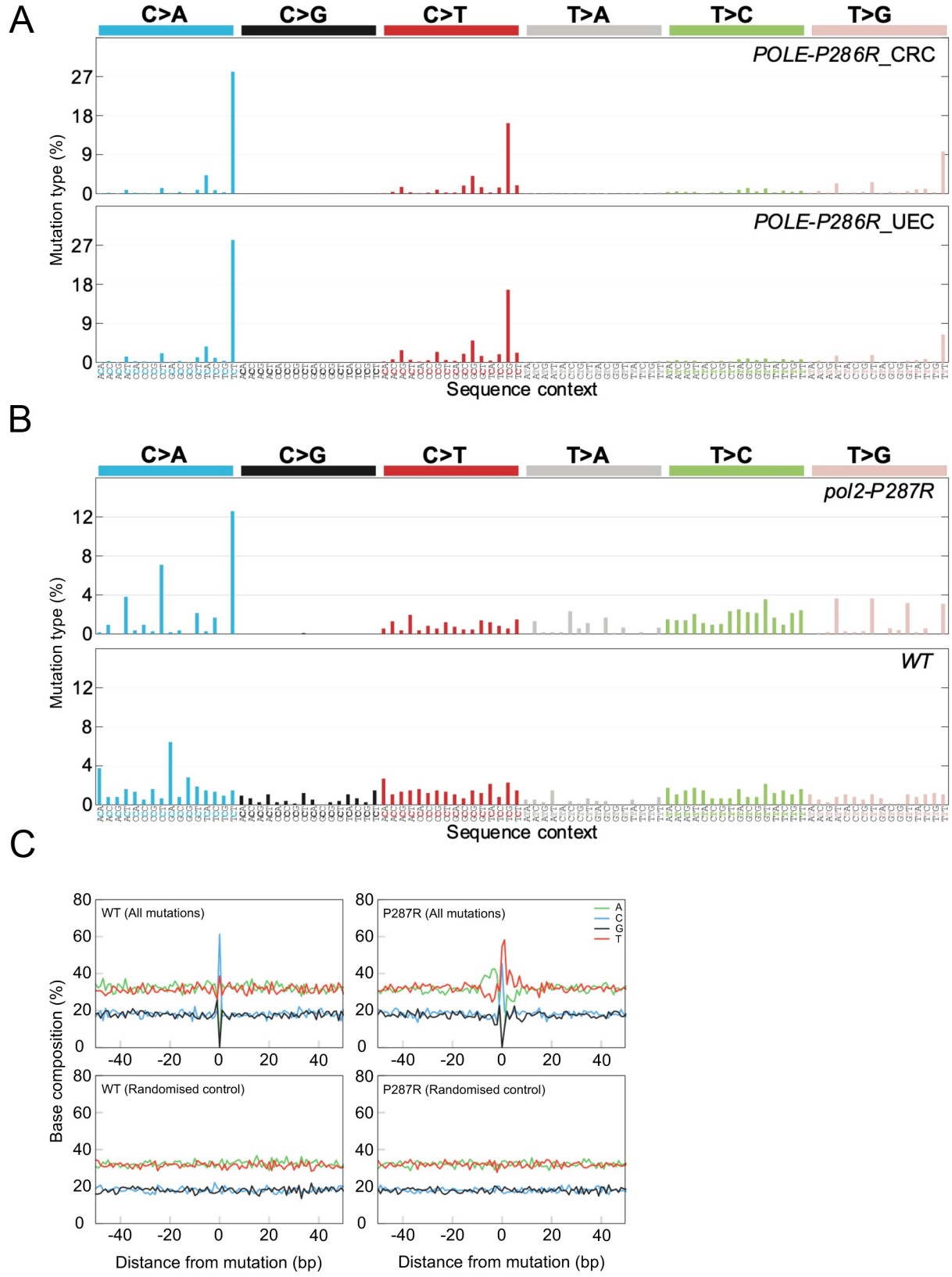

**Fig 2. Mutational spectra of the P286R variant in human adenocarcinomas and *S. pombe*.** (A) Mutation types in *POLE-P286R* colorectal carcinoma (top) and uterine endometrioid carcinoma (bottom). (B) Mutation types in *pol2-P287R* (top) and wild-type (bottom) [43] *S. pombe*. (C) Base composition in the vicinity of mutations in *pol2-P287R* and wild-type genomes.

## Sequence-context determinants affect mutation rate variability

To establish if mutations are generated in preferred nucleotide contexts, we aligned 103 bp long sequences, encompassing the trinucleotides harbouring the mutations and their 50 bp flanking regions, and determined the nucleotide frequency for each position relative to the centre of the alignment in both the wild-type and *pol2-P287R* cells. This showed 10–12 bp long region flanking the mutations in the case of the *pol2-P287R* strain with strong asymmetry in the frequency of adenine (A) and thymine (T) in the same DNA strand relative to the mismatch position (Fig 2C, top right). This asymmetric bias is absent in the region flanking wild-type mutations (Fig 2C, top left). As a control, alignment of randomly selected 103 bp sequences along the *S. pombe* genome generated flat profiles, in which the nucleotide composition coincided with the average genome content (Fig 2C, bottom panels).

Given the AT-asymmetry in the immediate vicinity of *pol2-P287R*-induced mutations, we looked at AT-rich regions in general to see if this sequence context is a factor in mutational probability in yeast and human genomes. This shows that in human genomes SNPs and TCT>TAT transversions in particular are enriched in AT-rich sequences <20 bp (S3 Fig and S6 Table). However, this bias is not seen in fission yeast *pol2-P287R*.

## Similarities between *pol2-P287R* mutational patterns and COSMIC signatures

To compare the mutational consequences of the human and yeast POLE286R variant, we first generated the mutational patterns of the wild-type and *pol2-P287R* strains and corrected for the difference in trinucleotide frequencies in the *S. pombe* genome and the human exome (Fig 3A), as described in [45]. Then, to assess if any of the mutational patterns determined in this study could be related to one of the COSMIC mutational signatures, a cosine heatmap was generated using the MutationalPatterns package [46]. A cosine similarity of 0.80 was considered a threshold for "high" similarity [45]. The *pol2-P287R* strain showed the highest similarity of 0.85 to SBS14 (Fig 3B and 3C and S3 Table). SBS14 was previously established as linked with concurrent *POLE* EDM and defective MMR [23–25]. The wild-type strain did not exhibit any cosine coefficient over the threshold, displaying a highest similarity of 0.73 to COSMIC signature SBS40 (Fig 3B and S3 Table). This is currently a signature of unknown aetiology, although numbers of mutations attributed to it are correlated with patients' ages for some types of human cancer. An analysis by signature decomposition yielded a similar result, with SBS14 (26%), SBS20 (16%), SBS10a (11%) and SBS35 (11%) contributing most to the *pol2-P287R* signature (S4 Table). None of these contributed to the pattern of wild-type mutations. These data indicate that the SBS10 associated with *POLE* EDMs in human cancers is not the most prevalent mutational pattern detected in the *pol2-P287R* strain. This may suggest that the mutational process in *S. pombe pol2-P287R* does not fully represent the situation in human cells. However, it is possible there could be differential efficiency of MMR in fungal and mammalian cells, and signature 10 could result from MMR-mediated correction of SBS14 introduced by *POLE* EDMs. In addition, human cancer cells are heterozygous for the *POLE-P286R* mutation, while the *pol2-P287R* mutation accumulation experiment used a haploid strain, so the mutagenic burden on mismatch repair would have been greater in the yeast strain.

## *pol2-P287R* cells exhibit increased sensitivity to DNA-damaging agents

Mutation of the N-terminus of Pol2 can lead to sensitivity to DNA damaging agents [47], possibly reflecting the role of Pol2 in various DNA repair pathways [3]. To investigate this, we performed spot assay analyses in the presence of different mutagenic agents. Results showed that

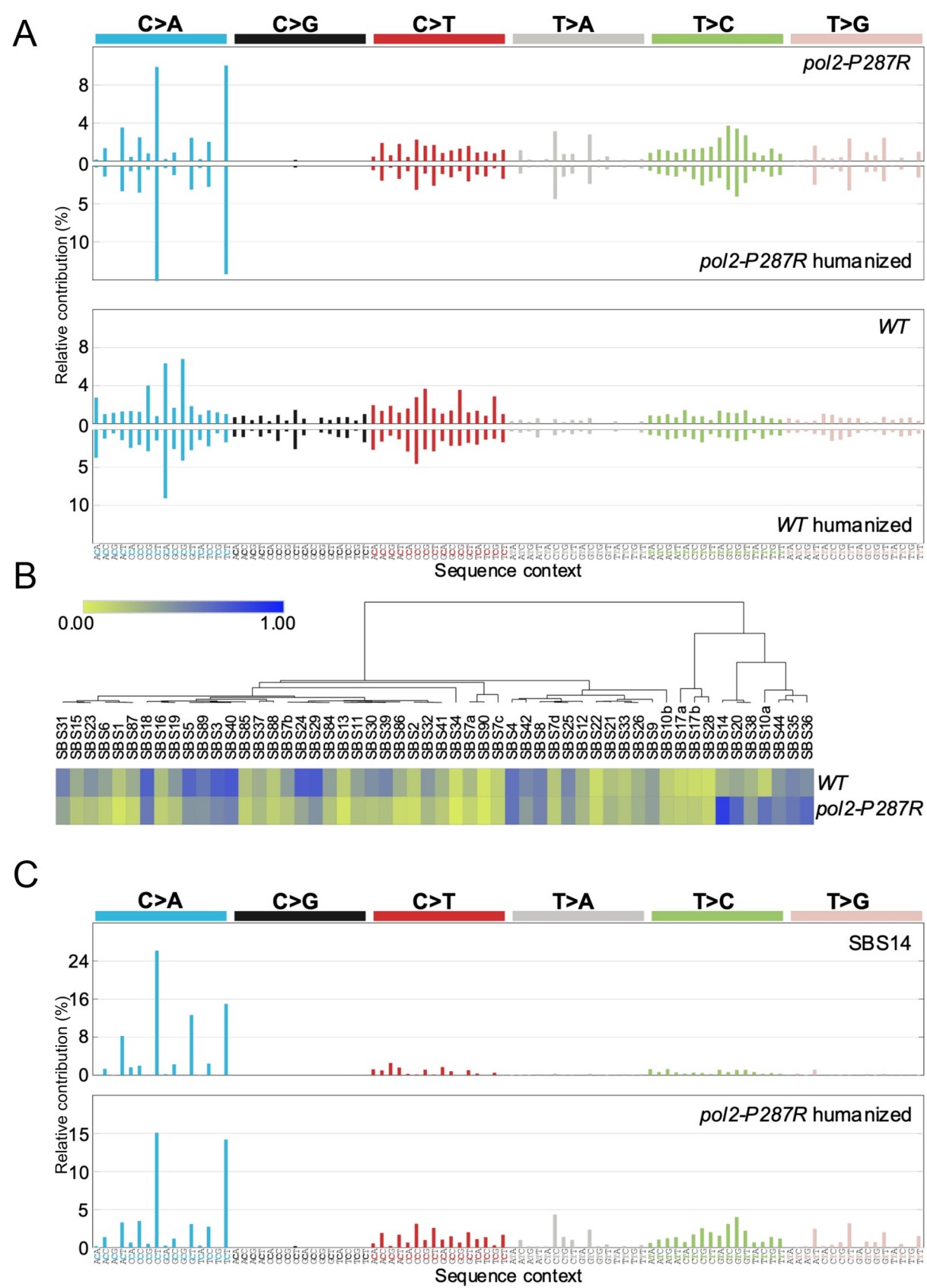

**Fig 3. Comparison between *S. pombe* mutational patterns and human cancer signatures.** (A) Base substitution patterns of *S. pombe* *pol2-P287R* and wild-type strains and their corresponding humanized versions (mirrored). (B) Heatmap of cosine similarities between the fission yeast mutational profiles and COSMIC signature. The signatures have been ordered according to hierarchical clustering using the cosine similarity between signatures, such that similar signatures are displayed close together. (C) Comparison between COSMIC signature 14 (SBS14) and *pol2-P287R* humanized profile.

*pol2-P287R* cells are very sensitive to agents inducing DNA breaks, such as bleocin, or nucleotide modification, such as MMS, 4-NQO, UV radiation, but not to HU, a ribonucleotide reductase inhibitor (Fig 4A). In fact, *pol2-P287R* cells showed some resistance to HU. The sensitivity to MMS is particularly dramatic and maybe due to the fact that, at the concentrations used, this alkylating agent generates 25 and 50-fold more lesions than 4-NQO and bleocin, respectively [48]. The sensitivity of *pol2-P287R* to DNA damaging agents is seen with several independently derived strains, so it is unlikely that this phenotype is due to second-site mutations facilitated by the strain's hypermutation. Given the sensitivity to a wide range of DNA damaging agents, it is possible that Pol2-P287R compromises the ability of the replication fork to cope with damaged template, or alternatively cellular responses to DNA damage may affect Pol2-P287R function (see Discussion).

## Increased dNTP levels lead to lethality in *pol2-P287R* cells

As described above, *pol2-P287R* cells exhibit slight resistance to HU (Fig 4A), suggesting that this mutant could have increased dNTP pools under normal conditions. Increased dNTP levels can elevate the mutation rate and, in combination with genetic alterations affecting DNA polymerase nucleotide selectivity, proofreading activity, or MMR, cause an enhanced mutator phenotype ([32,33,49] reviewed in [50]). However, the *pol2-P287R* strain showed similar levels of dNTPs to wild-type and *pol2-exonull* cells (Fig 4B), indicating that dNTP alterations are not contributing to its hypermutagenic phenotype. Similarly, elevated dNTP levels were not found in the *S. cerevisiae pol2-P301R* strain [29].

Increased dNTP levels are detrimental to the fidelity of DNA replication in bacteria, yeast and mammalian cells (reviewed in [50]). This effect might be exacerbated in the case of the error-prone Pol2-P287R variant. To address this possibility, we crossed a *cdc22-D57N* strain, where dNTP levels and mutation rates are enhanced due to inactivation of ribonucleotide reductase allosteric regulation [51], with the exonuclease-deficient *pol2-D276A/E278A* and *pol2-P287R* mutants and analysed the progeny by tetrad dissection. Double *pol2-D276A/E278A cdc22-D57N* mutants were readily obtained but the double *pol2-P287R cdc22-D57N* were generally not, implying synthetic lethality (Fig 4C). After extensive tetrad analysis however, we were able to obtain a double mutant, presumably harbouring a suppressor mutation. When compared with the single *pol2-P287R* mutant, the double mutant shows higher dNTP levels (S7 Table), an even higher mutation rate (S8 Table), and a growth defect (S4 Fig). Thus, it is likely that mutation burden of the *pol2-P287R* strain could be near the limit of the 'error-induced extinction', and an increase in the mutation frequency caused by raised dNTP levels is generally lethal but this is tolerated in the *pol2-D276A/E278A* control, which has a lower mutation rate.

## The *pol2-P287R* mutation causes S phase defects and is dependent on Rad3 for viability

We noticed that a proportion of *pol2-P287R* cells were elongated (S5A Fig), and the doubling time of the strain is longer than wild-type or exonull strains (Fig 5A), suggesting that problems with S phase might lead to a mitotic delay via checkpoint activation. To investigate this in

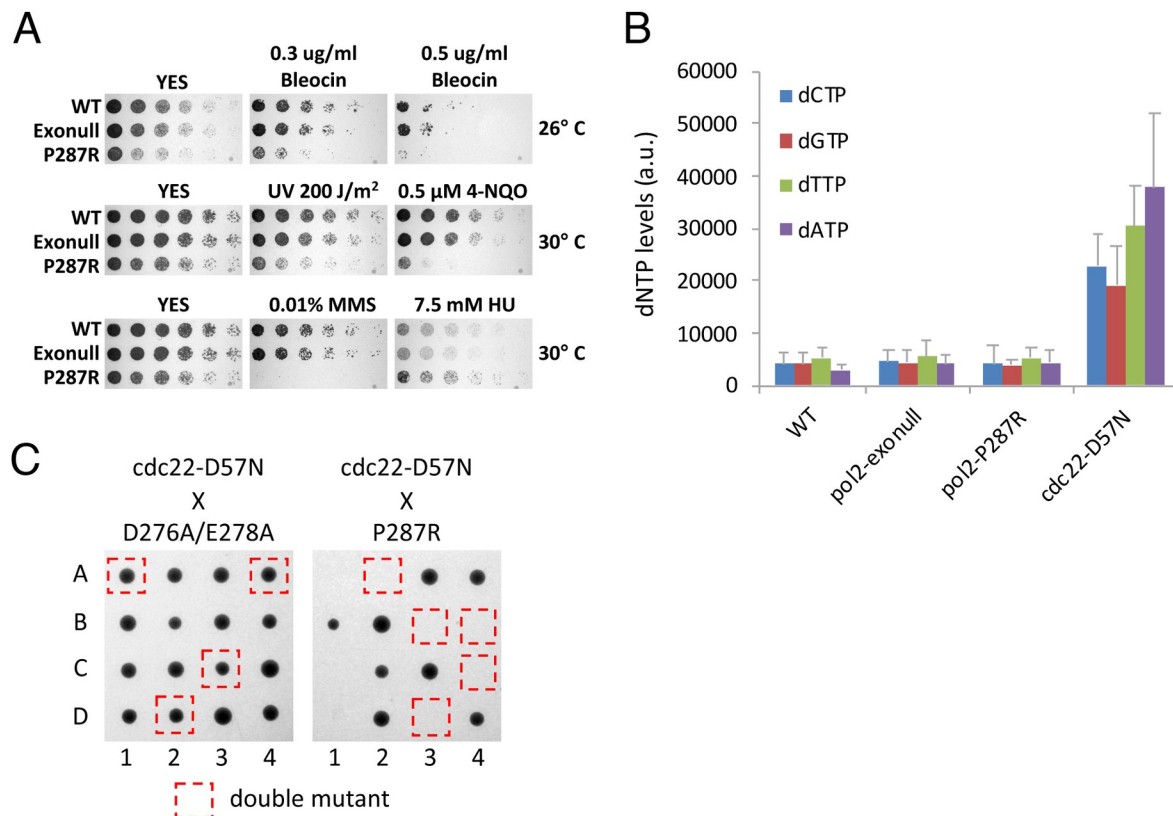

**Fig 4. The *pol2-P287R* strain is sensitive to DNA-damaging agents and increased dNTP levels.** (A) Spot assays to assess sensitivity to DNA-damaging agents of wild-type, exonull and *pol2-P287R* strains. 1:5 serial dilutions of the indicated strains were spotted on YES plates supplemented with 0.3 and 0.5 µg/ml Bleocin, 0.5 µM 4-NQO, 0.01% MMS, 7.5 mM HU or no drug and incubated at 26˚ or 30˚C for 2–3 days. (B) dNTP levels of the indicated strains measured from samples of exponential growing cells. Means ± SEs of three experiments are shown. The *cdc22-D57N* mutant, where allosteric regulation of RNR is inactivated, is shown as a positive control. Numerical data are in S7 Table. (C) Tetrad dissection of genetic crosses of between *cdc22-D57N* and *pol2* mutants. Representative spores from four asci are shown for each cross.

more detail, cells were arrested in G1 by nitrogen starvation, released from the block and flow cytometry was used to follow the progress of S phase. This showed a slower completion of S phase in the *pol2-P287R* strain (Fig 5B) compared to wild-type cells, while the wild-type and exonull strains showed similar S phase kinetics. To examine S phase execution by a different method, we pulsed cells with EdU for 15 minutes and measured the length of replication tracts by DNA combing (Fig 5C and 5D). The *pol2-P287R* cells showed shorter incorporation tracks, suggesting that fork progression is slower, or fork stalling is more frequent.

To determine if the Rad3 checkpoint kinase is required for viability of the *pol2-P287R* cells, we made double mutants with a temperature-sensitive *rad3ts* allele. While the exonuclease null double mutant was viable at the restrictive temperature, the *pol2-P287R* double mutant showed synthetic sickness (Fig 5E) and showed a high frequency of cut or enucleate cells (S5B Fig). While it was possible to derive *rad3Δ pol2-P287R* double mutants, these were slow growing (S5C Fig). We also found that foci of Rad11-GFP, a subunit of RPA, were more common in *pol2-P287R* cells compared to wild-type and exonull strains, suggesting that single-stranded DNA and perhaps DNA damage may result from replication by the mutant polymerase (Figs 5F and S5D). Consistent with these observations, we observed a low level of Chk1 and Cds1 phosphorylation in the *pol2-P287R* mutant (Fig 5G and 5H), and *chk1 pol2-P287R* or *cds1*

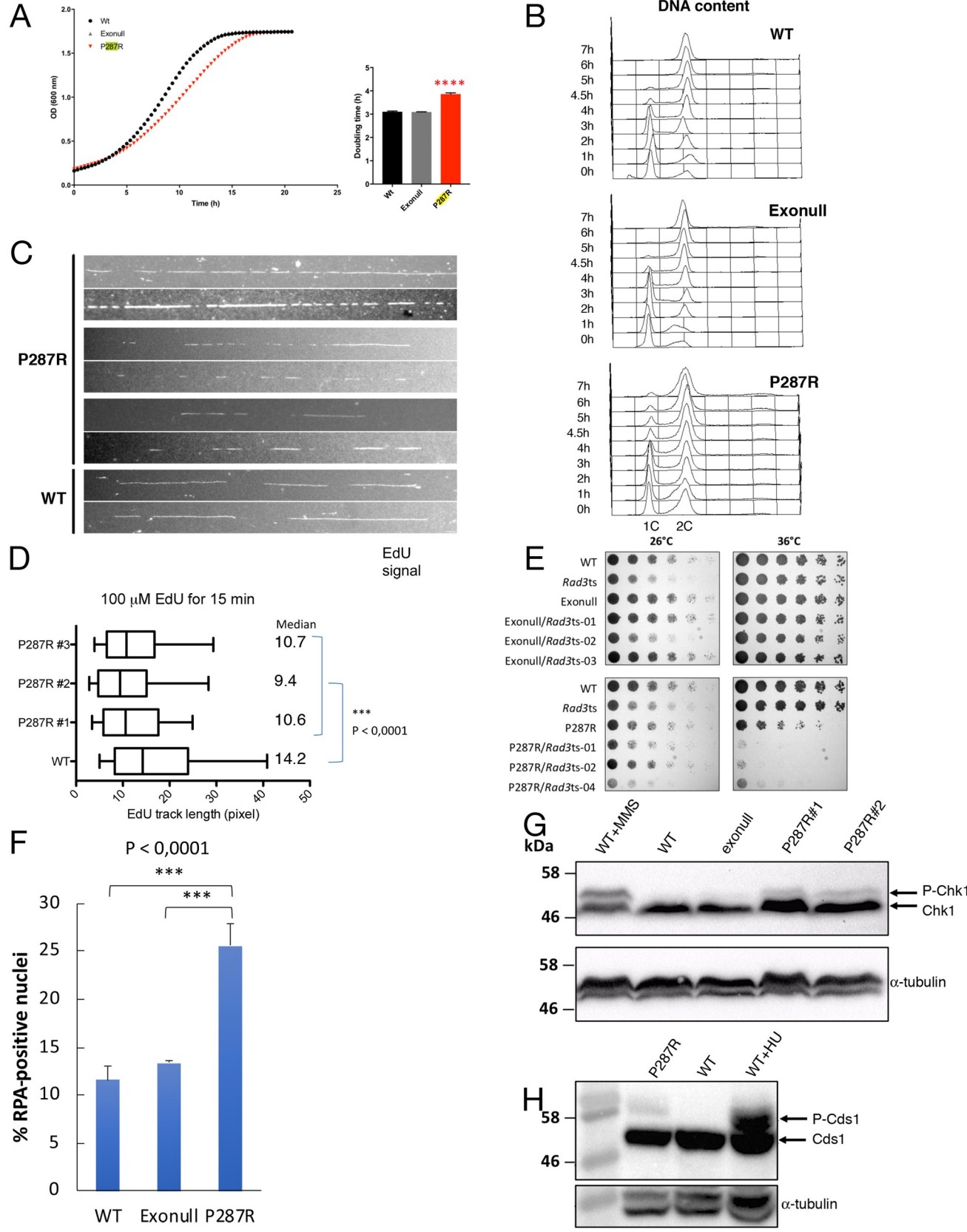

**Fig 5. *S. pombe pol2-P287R* cells show growth and S phase defects and DNA damage checkpoint activation.** (A) Growth rate of wild-type, exonuclease and *pol2-P287R* cells. Growth curves for the strains indicated were obtained by absorbance measurements every 20 minutes for 22h. The growth curves are representative of 3 independent experiments. **** P<0.0001. Numerical data are in S13 Table. (B) S phase progression in wild-type, exonull and *pol2-P287R* cells. Cells were arrested in G1 phase by nitrogen starvation, then refed and progress of S phase was monitored by flow cytometry. (C) DNA combing analysis of replication tracks in wild-type and *pol2-P287R S. pombe* cells. Strains used were modified to allow uptake of EdU. Cells were pulsed with 100 μM EdU for 15 min, then DNA was combed and EdU was detected using fluorescently-tagged AlexaFluor 488 Azide. (D) Quantitative analysis of track lengths from experiment described in (C). Numerical data are in S12 Table. (E) Spot testing of *rad3ts* strains. Serial dilutions of the indicated *S. pombe* strains were spotted to YES plates and incubated at the indicated temperatures. (F) Percentage of RPA foci in unstressed wild-type, exonull and *pol2-P287R* cells. Cells from the indicated strains were cultured to midlog phase in YES medium and imaged live. The numbers of foci in at least 350 nuclei were scored in three independent experiments, and mean values were plotted with error bars representing the standard deviation of the mean. *** P<0.0001 (Student's t-test). Numerical data are shown in S9 Table. (G) Analysis of Chk1 phosphorylation in wild-type, exonull and *pol2-P287R* cells. Chk1-HA strains were grown to log phase and protein extracts were analysed by western blotting. Wild-type cells grown in 0.025% MMS is shown as a positive control. α- tubulin was used as a loading control. (H) Analysis of Cds1-HA phosphorylation in unstressed wild-type and *pol2-P287R* cells. Wild-type cells grown in YES + 10 mM HU is shown as a positive control. Protein extracts were analysed by western blotting. α- tubulin was used as a loading control.

*pol2-P287R* double mutants were frequently slow growing (S5E Fig). Taken together, these observations suggest that a slower, defective S phase and accumulation of single-stranded DNA results from replication by Pol2-P287R, resulting in partial activation of the DNA damage and replication checkpoints.

## Genetic interaction with Pfh1 helicase

Given our observations that Pol2-P287R results in a defective S phase, we investigated whether there is a genetic interaction with Pif1/Pfh1 helicase, since this enzyme is involved in replicating through template barriers such as transcription complexes [52–54]. In *S. pombe*, Pfh1 has an essential role in the maintenance of both nuclear and mitochondrial DNA [55]. Therefore, to study its genetic interaction with the exonuclease variants, we made double mutants of *pol2-P287R* with a conditional allele of *pfh1 (nmt1-81Xpfh1-GFP)*, where the gene is under the control of a weak promoter that is down-regulated by thiamine [55]. In the absence of thiamine, Pfh1 is localized to the nucleus and mitochondria (Fig 6A, top panels) and all the strains grew well at 26˚, 30˚ and 36˚ C (Fig 6B, top panels). The nuclear signal is dramatically reduced in the presence of thiamine (Fig 6A, bottom panels). The growth of the *pol2-D276A/E278A nmt1-81Xpfh1-GFP* double mutant strain was similar to wild-type, when *pfh1-GFP* was repressed. However, *pol2-P287R nmt1-81Xpfh1-GFP* cells were very sick suggesting a strong genetic interaction with *pfh1* (Fig 6B, bottom panels).

It has been recently found that Pfh1 overexpression is able to rescue some replication defects in fission yeast [56]. We wondered whether Pol2-P287R defects could be partially alleviated by overexpressing this helicase. *pol2-287R* cells were transformed with pREP1 plasmid expressing Pfh1 DNA helicase under the control of the full strength (3X) *nmt1* promoter and spot assays were performed. Results indicated that overexpression of Pfh1 largely suppresses some defective phenotypes of *pol2-P287R* mutant cells, such as the growth defects at 30˚ C in minimal medium and the sensitivity to MMS (Fig 6C). However, the 4-NQO sensitivity was not rescued by the overexpression of the helicase and the mutation rate is enhanced (S11 Table). Overall, these results suggest that the replisome in *pol2-P287R* cells may have difficulty in dealing with 'hard-to-replicate' template particularly if Pfh1 levels are low, but enhanced Pfh1 activity may to some extent compensate for *pol2-P287R* defects.

## Pol κ deletion suppresses the hypermutagenic phenotype

In contrast to its strong mutator effect *in vivo*, it has been shown in *S. cerevisiae* that P286R does not have a high error rate *in vitro* [29]. We hypothesized that as a consequence of the failed proofreading caused by the *pol2-P287R* mutation, replication fork stalling might increase

A

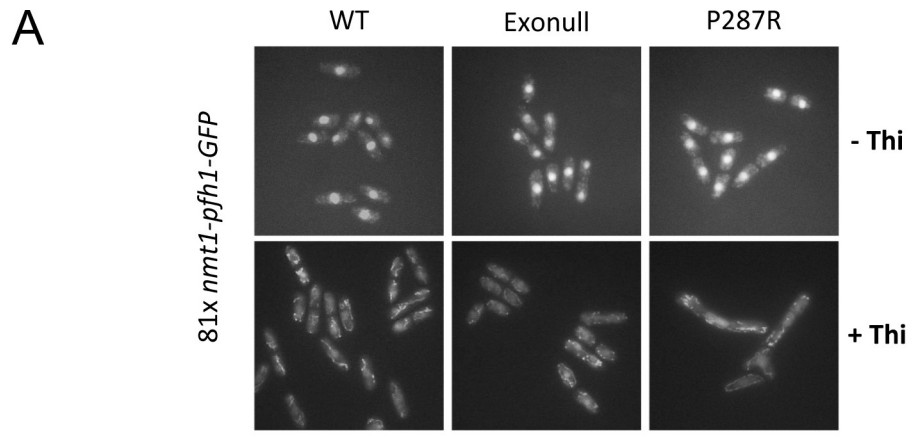

B

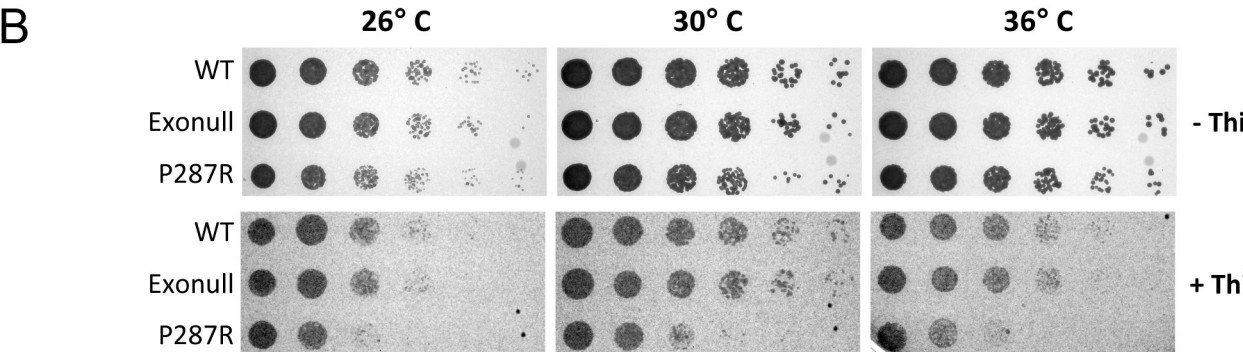

C

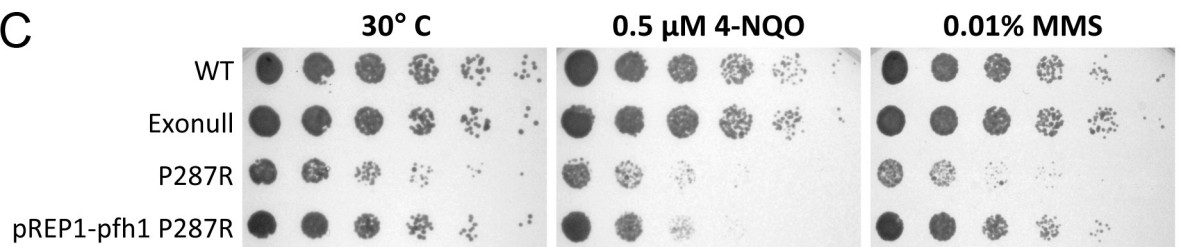

**Fig 6. Genetic interaction between *pol2-P287R* and *pfh1*.** (A) *nmt1-pfh1* cells were cultured to midlog phase in minimum medium containing 15 μM thiamine, or no thiamine and imaged live. Representative pictures under unrepressed (-Thi) or *pfh1* repression (+Thi) are shown. (B) 1:5 serial dilutions of the indicated strains with thiamine-regulatable *pfh1* (*nmt1-81Xpfh1-GFP*) were spotted onto minimum medium plus or minus thiamine and incubated 3 days at 26˚, 30˚ and 36˚C. (C) As (B) except cells were spotted onto minimal medium plates without thiamine containing 0.5 μM 4-NQO, 0.01% MMS or no drug.

and TLS polymerases could be recruited onto chromatin, to extend mismatched primer terminus, resulting in mutagenic synthesis [34]. We analysed the mutation rates of different TLS polymerase mutants containing a deletion of *rev1*, *rev3* (the catalytic domain of Pol ζ) or *kpa1* (Pol κ), or a single point mutation in *eso1* (*eso1-D147N*) inactivating Pol η, individually and in combination with *pol2-P287R*. Canavanine fluctuation assays showed that deletion of *rev3* in *pol2-P287R* cells did not significantly change the error rate (Fig 7A), consistent with data from *S. cerevisiae* [29]. In the case of *rev1Δ pol2-P287R* and *eso1-D147N pol2*-P287R double

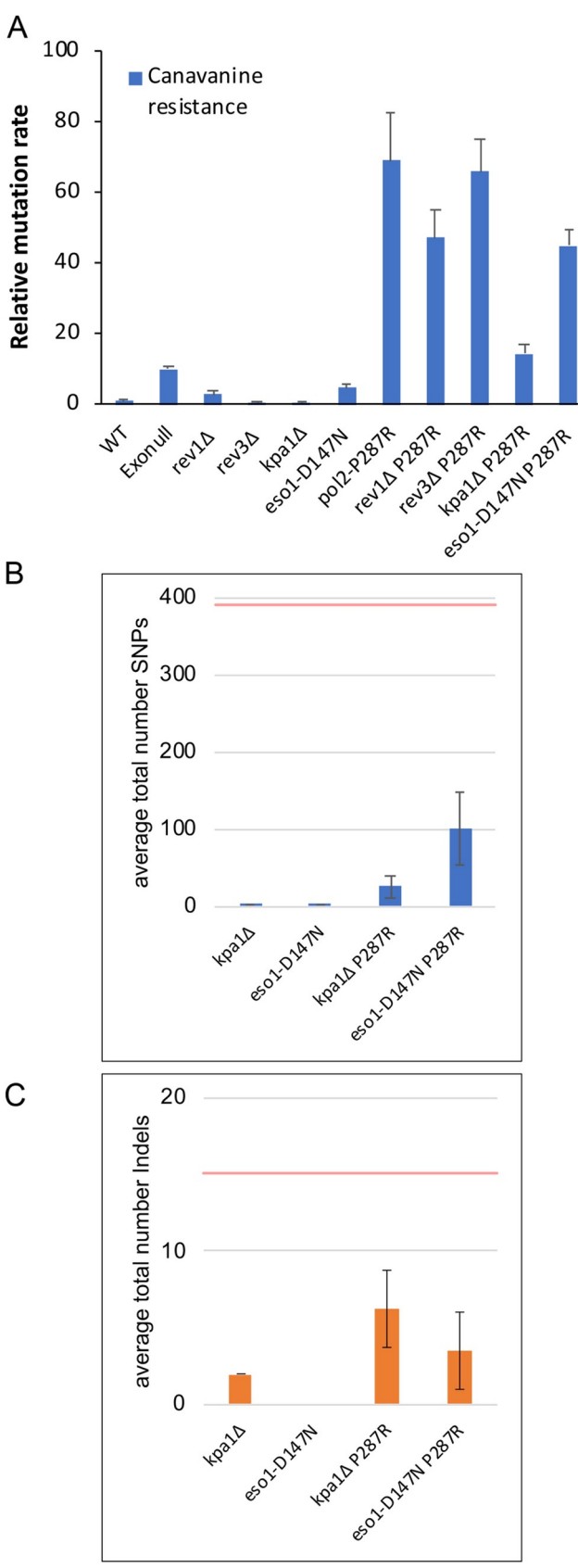

**Fig 7. Deletion of the TLS Pol κ and η genes reduces the mutation rate of *pol2-P287R* cells.** (A) Can$^R$ mutation rates of the indicated strains relative to wild type, determined by fluctuation analysis. Numerical data are in S13 Table. (B) Number of base substitutions during 200 generation MA experiments in single and double polymerase mutants. The horizontal red line shows the average number of SNPs in the *pol2-P287R* strain from Fig 1C. The number of lineages set up was: *kpa1Δ* 2; *eso1-D147N* 1; *pol2-P287R kpa1Δ* 3; *eso1-D147N pol2-P287R* 2. The average number of mutations is shown and error bars show the range of data. Numerical data are shown in S10 Table. (C) As (B) but number of indel mutations is shown. The horizontal red line shows the average number of indels in the *pol2-P287R* strain from Fig 1C.

mutants, the error rate was partially decreased (Fig 7A). Strikingly, *kpa1* deletion reduced the mutation rate of *pol2-P287R* to a level similar to an exonuclease deficient mutant, suggesting that this polymerase contributes to mutagenesis in cells expressing Pol2-P287R.

To confirm this result, we performed a ~200-generation mutation accumulation (MA) experiment followed by deep sequencing of genomic DNA from *kpa1Δ* and *eso1-D147N* single and double mutants. Sequencing data showed a reduction of around 4-fold in the number of mutations accumulated in the absence of Pol η activity compared to the single *pol2-P287R* mutant (Fig 7B and 7C). In accordance with the fluctuation rate assays, *kpa1Δ* caused a stronger reduction in the mutation burden, with a 15-fold decrease in the number of mutations accumulated the *kpa1Δ pol2-P287R* double mutant (Fig 7B and 7C). Overall, these results suggest that an important feature of hypermutation caused by the *pol2-P287R* variant is polymerase switching facilitated by TLS polymerases.

## Discussion

Defects in DNA replication have long been suggested to have a causative role in human cancer, but specific examples have only recently emerged. Recently, several studies show that somatic *POLE* proofreading domain mutations occur in sporadic ECs, CRCs and several other cancers (reviewed in [18,19]) The *POLE* EDMs are associated with a phenotype of hypermutation, microsatellite stability and favourable prognosis, possibly due to T cell responses to tumour neoantigens [57]. The most frequent variant, P286R, has been modelled in *S. cerevisiae* (Pol2-P301R), where it causes a strong mutator effect even in the heterozygous condition, in excess to that seen with exonuclease defective Pol2. [26]. However, the mechanisms leading to this hypermutation phenotype are not well understood.

Our results show that the equivalent mutation of *POLE-P286R*, *pol2-P287R*, is also highly mutagenic in fission yeast, with a base substitution frequency much higher than that caused by loss of the exonuclease activity of Pol ε. In consonance with findings in human cancers, the most abundant mutation is a C > A transversion in a TCT context, although the mutational pattern in yeast does not completely mirror the human pattern. The mutational pattern is not just a reflection of the polymerase defect, since it is also affected by mismatch repair, which may differ in its specificity and efficiency between yeast and humans. In addition, the *pol2-P287R* mutant shows reduced S phase kinetics, sensitivity to DNA damaging agents, increased RPA foci, dependence on Rad3, and a low level of Chk1 and Cds1 phosphorylation, indicating partial activation of the DNA damage and replication checkpoints. The human *POLE-P286R* mutation is found in the heterozygous condition, so we would not predict tumour cells carrying this mutation would necessarily show an S phase delay. Indeed, mice heterozygous for the P286R are born healthy and fertile, with no enhancement of DNA damage markers although they are highly cancer prone [58]. In contrast, mice hemizygous for the *POLE-P286R* mutation generally show embryonic lethality. Thus mammalian cells heterozygous for the *POLE-P286R* mutation may be ideally predisposed for cancer formation, having a sufficiently high mutation rate but without defects in cell cycle progression.

Budding yeast Pol2-P301R has a lower mutation rate than Pol2-exonull in vitro [29], so it is not clear what is responsible for the high *in vivo* mutation rate. One suggestion is that the strong mutator effect results from higher polymerase activity and more efficient mismatch extension capacity, which leads to misincorporation rather than proofreading [29]. Our data suggest that additional mechanisms are involved, namely that Pol κ, and to a lesser extent Pol η, contributes to the elevated mutation rate of the *pol2-P287R* mutant, similar to the situation seen with defective Pol α [34]. This situation may not be restricted to defective polymerases, as stalling of replication forks due to dNTP deprivation may also involve Pol κ [59,60]. Pol κ, and less so Pol η, are efficient at extending from a mismatch [61,62], and show error rates considerably higher than Pols δ and ε [62]. Previous work has suggested that following misincorporation, the budding yeast Pol2-P301R has a tendency to extend the mismatch [29], but the ability of Pol δ to proofread Pol2-P301R errors means that Pol ε must dissociate some of the time [37] possibly allowing extension by Pol κ. However, the notion that Pol κ 'writes' mutations in *pol2-P287R* is not consistent with the observation that *S. cerevisiae* Pol ε exonuclease domain mutants show the same SBS14 signature as in *pol2-P287R* [24], and *S. cerevisiae* does not have Pol κ. One possibility is that Pol κ (and η) may facilitate switching to Pol δ when the leading strand stalls (S6 Fig), since Pol δ has been shown to take over leading-strand synthesis after fork stalling [63]. Pol δ synthesis on the leading strand is known to be mutagenic [63,64] and could write SBS14. Interestingly, the next highest mutational signature in *pol2-P287R* and related *S. cerevisiae* mutants is SBS20, known to be associated with Pol δ mutations in the context of MMR inactivation [23]. Possibly MMR is not so efficient with DNA synthesised by Pol δ on the leading strand, as old/new strand discrimination requires ribonucleotide incorporation, which is much lower with Pol δ compared to Pol ε [65]. Interestingly, our data show that mutation of the proline equivalent to P286 in Pol δ does not lead to hypermutation, consistent with the fact that tumour sequencing has not identified this as a clinical mutation, and also the frequency of somatic Pol δ proofreading domain mutations seems to be much lower than Pol ε mutations [18]. This may reflect the fact that although switching to TLS polymerases can occur on the lagging strand, Pol δ carries out most synthesis, and efficient MMR may utilise nicks between Okazaki fragments rather than ribonucleotides for old/new strand discrimination [66], preventing hypermutation.

In addition to their roles in lesion bypass during translesion synthesis, Pols κ and η have important functions during the replication of hard-to-replicate sites such as microsatellite sequences, G-quadruplex and common fragile sites (CFSs), and during conditions of replicative stress [67]. Pol ε P287R may be more prone to stalling or displacement at such sequences compared to wild-type or exonull polymerases, as suggested by the interaction with Pfh1 reported here, perhaps again allowing polymerase switching.

*POLE* proofreading domain-mutant tumours may be particularly sensitive to specific therapeutic approaches due to their exceptional mutation burden. One strategy to decrease the overall fitness of hypermutating tumour cells is by increasing the mutation frequency further to induce an error catastrophe, similar to lethal mutagenesis of viruses [68]. Indeed, we found that *pol2-P287R* cells are sensitive to high dNTP levels, which probably reflects a further increase in mutation rate leading to cell death by error-induced catastrophe. The sensitivity of *pol2-P287R* cells to DNA damaging agents may also be due to the increased dNTP levels that result from DNA damage [69]. Normal somatic cells might be much less sensitive to a transient increase in mutation rate [68,70]. Clinically, an increase in the mutation rate could be effected by, for instance, decreasing the efficiency of MMR, or inducing DNA lesions using DNA damaging agents or nucleoside analogues. Additionally, upregulating RNR or inactivating SAMHD1 would also increase dNTP levels. Interestingly, SAMHD1 putative cancer drivers [71] are not found in any of the POLE-P286R tumours from a collection containing 2188

non-redundant samples from 10 colorectal and endometrial studies [72,73], suggesting that these mutations might be mutually exclusive.

In conclusion, while previous work has focused on features of POLE P286R function that are intrinsic to the enzyme, our results in *S. pombe* suggest that some aspects of the high mutation rate result from in vivo responses to polymerase malfunction. Clearly, it will be of interest to see a similar situation is also found in human cells.

## Materials and methods

### *S. pombe* methods

Standard media and genetic techniques were used as previously described [74]. Cultures were grown in either rich medium (YE3S), or Edinburgh minimal medium (EMM) supplemented with the appropriate requirements and incubated at 30˚C with shaking, unless otherwise stated. Nitrogen starvation was carried out using EMM lacking $NH_4Cl$. The relevant genotypes and source of the strains used in this study are listed in S1 Table. Oligos used for strain constructions are listed in S2 Table.

For spot testing, cells were grown overnight to exponential phase and five serial dilutions (1/4) of cells were spotted on agar plates using a replica plater (Sigma R2383). Plates were incubated at the relevant temperature, and photographed after 2–3 days.

Growth curves were obtained using the Bioscreen C MBR machine. Cultures in exponential phase were adjusted to $2 \times 10^6$ cells/ml and 250 µl were dispensed in 4–8 replicate wells for each strain. The plate was incubated at 30˚C with constant shaking at maximum amplitude. The OD600 of each well was measured every 20 minutes for 22 hours. Duplication time was determined via analysis of the exponential phase of the plotted log2 graph time vs OD600 using the exponential growth equation available on Prism.

Cell lengths were determined using an AxioPlan 2 microscope and pictures where taken using a Hamamatsu ORCA E camera system with Micro-Manager 1.3 software. Cell length was measured using ImageJ. Fluorescence microscopy was carried out as previously described [75]. For determining the percentage of cut or enucleate cells, cells were fixed with methanol and acetone and stained with DAPI (1 µg/ml) and methyl blue (50 µg/ml).

dNTP levels were determined as previously described [76].

*S. pombe pol2* variant strains were constructed as described previously [77]. The *pol2-P287R* strain was constructed using mutagenic primers 1128 and 1129; the P>R mutation generates an NruI site.

*S. pombe pol3* variant strains were constructed as previously described [78]. For strain *pol3P311R:kanMX6* (3419), mutagenic primers used were 1192 and 1193, with flanking primers 1148 and 1076. The mutation introduces a NruI site. The final PCR product was cloned into pFA6a as AscI-BamHI fragment and integrated at the *pol3* locus after cleavage with XhoI. The exonull strain *pol3D386A* was based on the mouse D400A mutation which has been shown to lack exonuclease activity. [79]. Mutagenic primers used were 1140 and 1141, with flanking primers 1075 and 1076; the mutation also introduces a NruI site. The final PCR product was cloned as above and integrated into the *pol3* locus after cleavage with CspCI.

### Protein analysis

Proteins were extracted by TCA method and resolved on 3–8% Tris Acetate NuPAGE (Thermo Scientific) in reducing conditions (NuPAGE Tris Acetate SDS running buffer–Thermo Scientific) [80]. For analysis of Chk1 and Cds1 phosphorylation, cells were grown to log phase in YES at 30˚C. As a positive control for Chk1 phosphorylation, cells were grown in YES plus 0.025% MMS. As a positive control for Cds1 phosphorylation, cells were grown in

YES plus 10 mM hydroxyurea. In the case of P-Chk1 analysis, proteins were resolved by SDS-PAGE using 10% gels with an acrylamide/bisacrylamide ratio of 99:1 [81]. Phos-tag acrylamide gels (Wako) were used to detect Cds1-P. Proteins were transferred onto a PVDF membrane by a dry transfer method using the iBlot2 Dry Blotting System (Thermo Fisher Scientific). Every step from blocking to washes to antibody incubation was performed via sequential lateral flow with the iBind Flex Western device. HA-tagged proteins were detected with the rabbit monoclonal antibody HA-Tag (C29F4). The secondary anti-rabbit used was Dako PO448. α tubulin was used as a loading control and the probing was done by using mouse monoclonal anti-α-tubulin (Sigma T5168). The secondary antibody used was goat anti-mouse (HRP) Ab97040.

## Mutation accumulation experiments

Strains were woken up from -70˚C, then single colonies were used to start lineages. One colony was grown up to prepare DNA for sequencing (generation 0) and restreaked on a fresh YES plate. After the colonies had grown up (3 days), a single colony was restreaked on a second YES plate. This process was repeated until the cells had gone through approximately 200 generations (11 passages). At the end of the experiment a single colony was grown up to prepare DNA for sequencing (generation 200). Variants were called (see sequencing analysis) and mutations were identified if present in generation 200 DNA but not in generation 0 DNA. Doubling time used to estimate the number of generations, as previously described [44].

## Mutation rate analysis by fluctuation analysis

Mutation to canavanine resistance was used to estimate mutation rates, as previously described (Kaur et al., 1999). A culture ($10^4$ cells/ml) was aliquoted into 12 wells of a 96-well microtitre plate (0.25 ml/well) and allowed to grow to saturation. Dilutions were plated onto YES plates to determine the cell concentration, and 0.15 ml of each culture was plated out onto a PMG plate (EMM-G, (Fantes and Creanor, 1984)) containing 80 μg/ml canavanine. Canavanine-resistant colonies were counted after 11 days at 30˚C. Mutation rates were calculated using the Ma-Sandri-Sarkar maximum likelihood estimator (Sarkar et al., 1992), implemented in rSalvador (Zheng, 2017).

## Sequencing analysis

Libraries were sequenced on a HiSeq2500 (Illumina) to generate 2x150 bases paired reads with an average genome coverage of ~200-fold and processed with RTA v1.18.66.3. FastQ files for each sample were obtained using bcl2fastq v2.20.0.422 software (Illumina). More than 6.5 M of pairs per sample were obtained. QC check of sequencing reads were revised with FastqC software and, then reads were trimmed off Illumina adapters using Cutadapt 1.16. Trimmed fastq file were aligned to *S. pombe* reference genome (ASM294v2.20 assembly) using Bowtie 2 with '—fr -X 1000 -I 0 -N 0—local -k 1' parameters. Sequencing duplicate read alignments were excluded using MarkDuplicates and sorted using Samtools. For bigwig (bw) representation and visual inspection bamCoverage and IGV software were used.

For the variant calling process, the bam files were processed with mpileup from Samtools and the calling process were executed with VarScan2. For somatic variant calling VarScan2 was used with both Generation 0 (G0) and Generation 200 (G200) mpileup file and only the High Confident (HC) variants were retained for next analysis. Human variant calling files from colorectal and endometrial POLE-P286R cancers were obtained from the cBioPortal curated set of non-redundant studies [72,73]. To calculate the consensus variants (both in *S. pombe* and human variants) of a genotype we retained SNPs from different genome positions

and, for the same position, only when the same change occurs in all variant files. All secondary analyses were made with custom python scripts available under request.

Sequencing data from this study have been deposited in the Gene Expression Omnibus (GEO) database under the accession number GSE169231 (https://www.ncbi.nlm.nih.gov/geo/query/acc.cgi?acc=GSE169231).

### Bioinformatic analysis

Genomic region annotation data (Fig 1D) were derived from the *S. pombe* reference genome (ASM294v2.20 assembly). The 'expected' values correspond to the percentage of each genomic region across the genome. To determine the 'observed' distribution, the number of substitutions at each genomic region were calculated using a custom Python script and percentages represented.

To establish the base substitution patterns of POLE-P286R colorectal and endometrial cancer, tsv files corresponding to CRC and UEC samples harbouring *POLE-P286R* mutation were downloaded from cBioportal [72,73] and the average percentage of single mutations across all individual samples per cancer type in their 5′ and 3′ base sequence context calculated and plotted using a custom Python script (Fig 2A). For the mutational patterns in fission yeast, 1073 and 746 single base substitutions from the P287R (this study) and WT [43,44] MA experiments respectively were identified and plotted as above (Fig 2B).

For determining base composition in the vicinity of mutations (Fig 2C) 103 bp long DNA sequences harbouring single base substitutions were aligned to the mismatched position. The percentage of mononucleotides were calculated for each position. Alignment of the same number of sequences 103-bp long randomly selected along the *S. pombe* genome were used as a control.

A + T content of regions harbouring single base substitutions in general, or TCT > TAT transversion in particular, ranging from 9–20 bp, 21–50 bp and >50 bp (S3 Fig) was calculated using *S. pombe* or human GRCh37 reference genomes for *pol2-P287R* and P286R-CRC/ P286R-UEC samples respectively. Expected values based on the genomic composition were also calculated as a control. To determine whether there was a statistically significant difference between the expected and the observed percentages, we used the chi-squared test.

Base substitution patterns of the wild-type and *pol2-P287R* strains (Fig 2B) were normalized by the trinucleotide frequencies in the fission yeast genome to generate the corresponding mutational patterns. The latter were then corrected for the difference in trinucleotide frequencies in the *S. pombe* genome and the human exome to establish the humanized versions (Fig 3A), as previously described [45].

Cosine similarity values for the comparison between humanized *S. pombe* mutation patterns with COSMIC v3.1 signatures (both adjusted to human whole-exome trinucleotide frequencies) were calculated using the Python cosine_similarity script from Scikit-learn [82].The heat map (Fig 3B) was generated using Morpheus software (Morpheus, https://software.broadinstitute.org/morpheus)

The contribution of COSMIC v3.1 signatures to the humanized fission yeast mutation patterns was determined using Non-negative Matrix Factorization (NMF). The calculated weights for each signature were subsequently converted into the equivalent percentage for the representation.

### Flow cytometry

Cells were fixed in 70% ethanol and analysed after RNase digestion and SYTOX Green staining as previously described [83].

## DNA combing

DNA combing was carried out basically as previously described [84,85], using a Genomic Vision Molecular Combing system and CombiCoverslips. A pulse of 100 μM EdU for 15 minutes was used to label DNA and incorporated EdU was detected using AlexaFluor 488 Azide A10266 (ThermoFisher Scientific) A10266.

## Supporting information

**S1 Fig. Steady-state levels of Pol ε variants in *S. pombe*.** Epitope-tagged polymerases were detected using anti-HA monoclonal antibody. α-tubulin is shown as a loading control.
(TIF)

**S2 Fig. Percentage and type of base substitutions in *S. pombe* expressing wild-type [43] and P287R Pol ε. Numerical data are in S13 Table.**
(TIF)

**S3 Fig. Observed and expected frequency of all SNPs and TCT>TAT transversions in *S. pombe* and human genomes.** Corresponding numerical data is shown in S6 Table. Differences from expected frequency is significant for human 9–20 bp AT-rich sequences (\*\*\* p<0.01; \*\* p<0.05).
(TIF)

**S4 Fig. Spot tests comparing the growth rate of *cdc22-D57N pol2-P287R* strain with parental strains.**
(TIF)

**S5 Fig. Genetic interactions between *pol2-P287R* and *rad3*, *cds1* and *chk1*.** (A) Cell length measurements of *S. pombe* cells in log phase. 200–400 cells were measured for each strain. Statistical analysis used a Krustal-Wallis test (\*\*\* P<0.001). "Wt construct" was constructed in the same way as the other two strains except that *pol2* mutations were not introduced. (B) Percentage of cut or enucleate cells in *rad3ts pol2-P287R* and parental strains. Log phase cells grown at 26˚C were shifted to 36˚C for 3 hours before examination by fluorescence microscopy. (D) Rad11-GFP (RPA) foci in live unstressed cells. Numerical data are shown in S9 Table. (E) Tetrads from *cds1Δ* or *chk1Δ* x *pol2-P287R* crosses. Blue triangles indicate double mutants.
(TIF)

**S6 Fig. Model for hypermutagenesis by Pol2P287R.** In contrast to proofreading by wt Pol ε (A), Pol2P287R can more readily extend from a mismatch (B, upper arrow). TLS polymerases (κ, η) may however facilitate switching to Pol δ (B, lower arrow). Synthesis by TLS polymerases is error prone, and DNA synthesised by Pol δ after switching may show more mutations, possibly due to defective MMR. In the absence of TLS polymerases, switching to Pol δ may be less efficient, allowing continued Pol ε synthesis. Switching or misincorporation may also result in uncoupling from the CMG and Rad3 activation. See text for further details.
(TIF)

**S1 Table. *S. pombe* strains used in this study.**
(DOCX)

**S2 Table. Oligos used in this study.**
(DOCX)

**S3 Table. Cosine coefficients between humanized *S. pombe* mutational patterns and COS-MIC v3.1 SBS signatures.**
(DOCX)

**S4 Table. Contribution of COSMIC Single Base Substitution (SBS) signatures to *S. pombe* mutational patterns.**
(DOCX)

**S5 Table. Comparison between *S. pombe* mutational patterns and *POLE* human cancer signatures.**
(DOCX)

**S6 Table. Percentage of mutations in AT-rich sequences.**
(DOCX)

**S7 Table. dNTP levels (arbitrary units) of *S. pombe* strains.**
(DOCX)

**S8 Table. Quantitative impact of increased dNTP levels on *pol2-P287R* mutagenesis.**
(DOCX)

**S9 Table. Rad11(RPA)-GFP foci numerical data.**
(XLSX)

**S10 Table. SNP and Indel numerical data from mutation accumulation experiments.**
(XLSX)

**S11 Table. Mutation rate of the *pol2P287R* strain over-expressing Pfh1.**
(DOCX)

**S12 Table. Numerical data for Fig 5D.**
(XLSX)

**S13 Table. Numerical data for Figs 1B, 5B, 7A, S2, S5A and S5B.**
(XLSX)

## Acknowledgments

We are grateful to Catherine Green and Jane Mellor for providing lab facilities following the abrupt closure of the Tinbergen building in February 2017. We thank Tony Carr, Stéphane Coulon, Christian Holmberg, Edgar Hartsuiker, Paul Russell, Virginia Zakian, Yolanda Sanchez and NBRP Japan for strains or plasmids. We are grateful to Philippe Pasero and Julie Saksouk for providing facilities and guidance regarding DNA combing.

## Author Contributions

**Conceptualization:** Ian Tomlinson, Sue Cotterill, Stephen E. Kearsey.

**Data curation:** Ignacio Soriano, Enrique Vazquez.

**Formal analysis:** Ignacio Soriano, Enrique Vazquez, Ellen Heitzer.

**Funding acquisition:** Ian Tomlinson, Sue Cotterill, Stephen E. Kearsey.

**Investigation:** Ignacio Soriano, Nagore De Leon, Sibyl Bertrand, Ellen Heitzer, Sophia Toumazou, Zhihan Bo, Chen-Chun Pai.

**Methodology:** Ignacio Soriano, Nagore De Leon, Stephen E. Kearsey.

**Project administration:** Stephen E. Kearsey.

**Resources:** Ignacio Soriano, Enrique Vazquez, Claire Palles.

**Software:** Enrique Vazquez.

**Supervision:** Timothy C. Humphrey, Stephen E. Kearsey.

**Validation:** Ignacio Soriano, Sibyl Bertrand.

**Visualization:** Ignacio Soriano, Stephen E. Kearsey.

**Writing – original draft:** Ignacio Soriano.

**Writing – review & editing:** Stephen E. Kearsey.

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
