## [Decision Letter · Decision Letter 0]

27 Apr 2021

Dear Dr Kearsey:

Thank you very much for submitting your Research Article entitled 'Expression of the cancer-associated DNA polymerase ε P286R  in fission yeast leads to translesion synthesis polymerase dependent hypermutation and defective DNA replication' to PLOS Genetics.

The manuscript was fully evaluated at the editorial level and by independent peer reviewers. The reviewers appreciated the attention to an important topic but identified some concerns that we ask you address in a revised manuscript

We therefore ask you to modify the manuscript according to the review recommendations. Your revisions should address the specific points made by each reviewer.

It is noted that some reviewers made comments that may require additional experimentation. In those cases, please clearly indicate whether you consider the additional experiments are needed to fully address the reviewer's concerns. We would be willing to allow a longer response time if you deem additional experiments are needed.

[LINK]

Yours sincerely,

John L. Nitiss

Guest Editor

PLOS Genetics

Peter McKinnon

Section Editor: Cancer Genetics

PLOS Genetics

Reviewer's Responses to Questions

**Comments to the Authors:**

Reviewer #1: The POLE P286R mutation is strongly associated with a hypermutation phenotype in human cancers. The precise reason why POLE P286R causes this phenotype is unknown. A number of previous studies have attempted to address this using various model systems include human, mouse and budding yeast, however, the question remains not fully resolved. The authors describe the use of a fission yeast model of the POLE P286R mutation. They show that the mutations generated in these mutants largely recapitulate human tumours (except the lack of mCpG in yeast). By testing the impact of different DNA damaging agents on the POLE mutant yeast, they conclude that these cells are highly sensitive to increase dNTP levels. Most significantly, they show that deletion of pol kappa, and to some extent pol eta, rescues the hypermutation phenotype, suggesting that the increased usage of these error-prone polymerases may be the reason why POLE mutants are hypermutated. Although these observations are not tested independently in other systems, nevertheless, it is an important contribution to our knowledge of POLE hypermutation. I only have some relatively minor comments for the authors, which are listed below:

1. The expected mutation rate should take in to account the GC content of the different regions. Ideally, something like SigProfilerSimulator (https://bmcbioinformatics.biomedcentral.com/articles/10.1186/s12859-020-03772-3) should be used to simulate the expected mutations and to account for trinucleotide content of the different regions.

2. The comparison of % mutations in AT-rich regions should be more quantitative between pombe and humans. The % should be normalised by the size of the respective AT-rich regions. For instance, the fact that pombe has a higher proportion of the genome with >50bp rich should be accounted for.

3. For the comparison of pombe and human P286R mutational signature, apart from using cosine similarity, it would be better to perform signature decomposition to get a more precise contribution of each signature in the pombe mutant.

4. “Cells with a high rate of spontaneous mutations frequently display increased sensitivity to DNA damaging agents [43, 44].” I don’t think this statement is necessarily true. For example, mismatch repair deficient cells (which have a high rate of spontaneous mutations) are resistant to temozolomide (https://www.ncbi.nlm.nih.gov/pmc/articles/PMC3494444/). The statement should be revised to perhaps specifically refer to DNA POLE mutants?

5. When comparing the number of mutations with WT versus the kpa1 and eso1 mutants (Fig 7B), it is confusing to show the indels on the same plot because the reference mutation number of P287R alone is presumably just SNPs? It would be better for indels to be plotted separately. Furthermore, were there any replicates in these experiments? Error bars would be helpful.

Reviewer #2: In this manuscript, Soriano et al. present an analysis of the fission yeast DNA polymerase epsilon mutant P286R, orthologue of the corresponding POLE mutant identified as driver of hypermutated colorectal and endometrial human cancers. They show that these mutants show mutagenic phenotypes comparable to those of the corresponding human or S. cerevisiae pol epsilon mutants. Finally, they show that this mutant is hypersensitive to DNA damaging agents, that this sensitivity is modulated by the helicase PIF1 and that a large fraction of the mutagenic phenotype is due to the presence of trans-lesion DNA polymerase kappa.

The manuscript is largely well written, interesting, and the experiments have been appropriately designed. However, the bioinformatic analysis of mutation accumulation experiments has not been carried out to a sufficient depth, especially given that the mutations introduced by a strand-specific DNA polymerase are likely to present a positional bias associated with DNA replication origins (indeed it is the case for S. cerevisiae and mammalian cells). The most striking finding of this manuscript, the substantial suppression of the mutagenic phenotype of P286R cells by inactivation of pol kappa is relegated to the final figure and not well investigated or discussed.

I believe this manuscript could be improved by addressing the following points:

1. Page 4 line 101. The discussion of signatures SBS10a and SBS10b does not take into account that by now several authors in different organisms have shown that the signature produced by pol epsilon EDM mutants is SBS14 (Haradhvala et al., 2018; Tracy et al., 2020; Herzog et al.,2021). SBS10a/b, which are observed in some human cancers with pol epsilon EDM alleles (but not in the yeast counterparts) might instead originate from MMR-mediated skewing of SBS14 after it was introduced at the replisome of pol epsilon EDM cells.

2. Page 5 line 112. Describing the “mutator effect” of pol2-P301R as “comparable with complete MMR deficiency” is potentially misleading, since the authors are clearly only referring to the mutation (accumulation?) rate and not to the mutational patterns. I suggest that the actual figures/rates are cited here.

3. Page 5 line 130-31. It has been shown that pol2-P301R does not lead to increased dNTP levels (Xing et al. 2019).

4. Page 6 line 156-164. What does “on average” refer to (“On average, we identified 0.5, 9.5 and 394 single-base substitutions in…”)? Is it the average between different mutation accumulation lines of the same genotype? If so, please add standard deviation whiskers and/or plot every single MA line in fig 1C to show the dispersion around the average. What is the total number of variants studied in each genotype? Is it sufficient for the subsequent signature analyses?

Also, there seems to be a small increase in INDELS in P287R compared to the wild-type (a log scale, or separating INDELS and SBS in two different graphs would make it clearer). Is it statistically significant? It was shown recently that S. cerevisiae EDM mutants generate a large amount of insertions, which are almost completely masked by MMR, but can still be detected in pol2-P301R cells (Herzog et al, 2021).

5. Page 7 line 172. The authors should specify that this refers to the mutation spectrum of EDM alleles after it has been altered by MMR and possibly pol delta proofreading activity (Flood et al. 2015; Bulock et al. 2020).

6. Page 7 line 173-184: The authors should include a synthetic measure of the similarity of the different mutational patterns shown in fig 2A-B, such as cosine similarity after normalizing for the trinucleotide composition of each organism.

7. Page 7 line 185-192. When discussing the nature of the mutational patterns observed in P287R cells, it should be noted that pol epsilon mutants only introduce mutations on the leading strand on which they operate. Therefore, for a meaningful discussion of the frequency of different mutational contexts, these should be reverse complemented when the corresponding mutations found on regions replicated as lagging strand (i.e. left of replication origins) to reflect their likely leading strand origin.

8. Page 8 lines 204-212. The rationale for this analysis is not well presented; and while this reviewer agrees with the general conclusion (“the mutagenic mechanism due to pol2-P287R is distinct from that seen in wild-type cells”), this conclusion does not seem to follow from the data presented in this paragraph and fig. 2D. It is also not well explained why the authors are focusing on the single TCT>TAT mutation and not on all the observed mutations, and it is not clear what one should conclude from the results presented.

9. Page 8 line 227. The fact that that the mutational pattern of pol2-P287R has similarity with SBS14 even in the presence of functional MMR is not unexpected. This could reflect a differential efficiency of MMR-mediated alteration of SBS14 in fungi and mammalian cells. On a technical point, the low resolution of Figure 3B makes it difficult to judge the colour of adjacent signatures in this version of the pdf.

10. Page 9 line 251. See point 3. The equivalent mutation in S. cerevisiae has similarly been shown not to lead to increased dNTP levels (Xing et al. 2019).

11. Page10 Line 263. Given the similarity of the mutagenic mechanism of pol2-exo– and pol2-P287R, it would have been extremely interesting to show the mutational pattern in pol2-D276A/E278A cdc22-D57N (the only viable pol2 mutant) cells to determine the quantitative impact of increased dNTP levels on mutagenesis in different sequence contexts. The authors should at least show that this double mutant has an increased mutation rate compared to the single mutants by means of fluctuation tests.

12. Page 11 line 302. Does Pfh1 overexpression affect the mutation rate of pol2-P287R?

13. The substantial reduction in mutation rates of pol2-P287R observed with pol kappa mutants and to a lesser extent with pol eta mutants is extremely interesting, however several points could be addressed in more depth with experiments and/or discussion. For example:

a. Does pol2-P287R pol kappa pol eta triple mutant further reduce the mutation rate?

b. Is there any synthetic lethality in the triple mutant? If not, one should assume that other more accurate polymerases, perhaps pol delta, take over from stalled pol epsilon?

c. If a large part of the mutations were indeed physically introduced by pol kappa in pol2-P287R cells, one would expect pol kappa to be responsible for “writing” SBS14, however SBS14 has been detected also in S. cerevisiae pol2-P301R MMR— strains, an organism that does not have a pol kappa homolog.

Reviewer #3: This MS from Steven Kearsey’s group examines the consequences of introducing the cancer prone POLε-P286R mutant into the fission yeast genome (the position is conserved as residue 287). This causes an increased mutation rate, and a mutation spectrum very similar to the one reported in human cancers. Interestingly, the mutations are suppressed by inactivation of translesion polymerases, suggesting that polymerase switching is involved. Consistent with this, it is shown by DNA combing and FACS analysis that replication proceeds slowly in the mutant, which is also synthetic lethal with absences of the Rad3 checkpoint. Finally, overexpression of the Pfh1 helicase can suppress some of the phenotypes associated with fission yeast P287R, whereas its deletion enhanced the phenotype.

Although I find this study interesting and compelling, I think there are some major problem with it in its present form. The work is manly confirmatory to previous studies in humans and budding yeast, but the potentially very interesting findings (see point 2 and 3 below) are not really pursued, despite the fact that fission yeast genetics should be most helpful in this process.

1. Although the mutation analysis is compelling, I missed a detailed discussion of the experimental setup chosen. Why was the cultures grown for 200 generations? What is the detection threshold for variant calling by the sequencing protocol? Were the identified mutants fixed in the population? Is phenotypic selection important in this time scale?

2. In my opinion, the most interesting aspect of the paper is the reported co-lethality with the cdc22-D57N mutant defective in feedback inhibition of ribonucleotide reductase (RNR). The authors ascribe this to the combined mutation burden of the two mutagenic strain, but provide no evidence for this. However, cdc22-D57N can readily be combined with ddb1 and this gives rise to a strong mutagenic synergy; yet the double mutant is alive (Fleck et al 2013). Is the lethality suppressed by loss of TLS polymerases, which one would expect if the mutation burden were the problem? The co-lethality with cdc22-D57N should be pursued – for instance by a suppressor screen, or perhaps by testing if the D57N effect is dominant in a heterozygous context.

3. It is also very interesting that P287R is synthetically lethal with rad3, but I find it hard to believe that the observed weak activation of Cds1 and Chk1 can explain this. Again, a genetic analysis should be very informative.

4. Why does the P287R strain have a much more severe phenotype than the strain lacking proofreading altogether? What is happening at the fork that causes checkpoint activation and polymerase switching in this mutant?

Minor comments:

L205: Delete “to see”

L245: an effect on -> effects

**Have all data underlying the figures and results presented in the manuscript been provided?**

Reviewer #1: Yes

Reviewer #2: Yes

Reviewer #3: Yes

PLOS authors have the option to publish the peer review history of their article (what does this mean?). If published, this will include your full peer review and any attached files.

Reviewer #1: **Yes: **Jason Wong

Reviewer #2: No

Reviewer #3: **Yes: **Olaf Nielsen

---

## [Editor Report · Decision Letter 1]

11 Jun 2021

Dear Dr Kearsey,

We are pleased to inform you that your manuscript entitled "Expression of the cancer-associated DNA polymerase ε P286R  in fission yeast leads to translesion synthesis polymerase dependent hypermutation and defective DNA replication" has been editorially accepted for publication in PLOS Genetics. Congratulations!

Yours sincerely,

John L. Nitiss

Guest Editor

PLOS Genetics

Peter McKinnon

Section Editor: Cancer Genetics

PLOS Genetics

Comments from the reviewers (if applicable):

**Data Deposition**

http://datadryad.org/submit?journalID=pgenetics&manu=PGENETICS-D-21-00425R1

**Press Queries**

---

## [Editor Report · Acceptance letter]

29 Jun 2021

PGENETICS-D-21-00425R1 

Expression of the cancer-associated DNA polymerase ε P286R  in fission yeast leads to translesion synthesis polymerase dependent hypermutation and defective DNA replication 

Dear Dr Kearsey, 

We are pleased to inform you that your manuscript entitled "Expression of the cancer-associated DNA polymerase ε P286R  in fission yeast leads to translesion synthesis polymerase dependent hypermutation and defective DNA replication" has been formally accepted for publication in PLOS Genetics! Your manuscript is now with our production department and you will be notified of the publication date in due course.

With kind regards,

Katalin Szabo

PLOS Genetics

On behalf of:
